# Weibull-Type Incubation Period and Time of Exposure Using *γ*-Divergence

**DOI:** 10.3390/e27030321

**Published:** 2025-03-19

**Authors:** Daisuke Yoneoka, Takayuki Kawashima, Yuta Tanoue, Shuhei Nomura, Akifumi Eguchi

**Affiliations:** 1Center for Surveillance, Immunization, and Epidemiologic Research, National Institute of Infectious Diseases, Tokyo 162-8640, Japan; 2Department of Mathematical and Computing Science, School of Computing, Institute of Science Tokyo, Tokyo 152-8552, Japan; kawashima@c.titech.ac.jp; 3Faculty of Marine Technology, Tokyo University of Marine Science and Technology, Tokyo 135-8533, Japan; ytan001@kaiyodai.ac.jp; 4Department of Health Policy and Management, School of Medicine, Keio University, Tokyo 160-8582, Japan; nom3.shu@gmail.com; 5Center for Preventive Medical Sciences, Chiba University, Chiba 263-8522, Japan; siero5335@gmail.com

**Keywords:** three-parameter Weibull distribution, γ-divergence, exposure time, infectious disease, robust estimation

## Abstract

Accurately determining the exposure time to an infectious pathogen, together with the corresponding incubation period, is vital for identifying infection sources and implementing targeted public health interventions. However, real-world outbreak data often include outliers—namely, tertiary or subsequent infection cases not directly linked to the initial source—that complicate the estimation of exposure time. To address this challenge, we introduce a robust estimation framework based on a three-parameter Weibull distribution in which the location parameter naturally corresponds to the unknown exposure time. Our method employs a γ-divergence criterion—a robust generalization of the standard cross-entropy criterion—optimized via a tailored majorization–minimization (MM) algorithm designed to guarantee a monotonic decrease in the objective function despite the non-convexity typically present in robust formulations. Extensive Monte Carlo simulations demonstrate that our approach outperforms conventional estimation methods in terms of bias and mean squared error as well as in estimating the incubation period. Moreover, applications to real-world surveillance data on COVID-19 illustrate the practical advantages of the proposed method. These findings highlight the method’s robustness and efficiency in scenarios where data contamination from secondary or tertiary infections is common, showing its potential value for early outbreak detection and rapid epidemiological response.

## 1. Introduction

Identifying the precise time of exposure to a newly emerging infectious pathogen using symptom onset data is a fundamental step for locating infection sources and implementing effective public health measures such as contact tracing, particularly in diseases transmissible via human-to-human contact [1,2]. Once the exposure time is well estimated, one can also determine the incubation period, which is defined as the time from the exposure event to symptom onset [3]. Historically, a variety of parametric distributions including exponential, lognormal, and Weibull distributions have been employed to model this period [4,5]. Nishiura (2007) provides a comprehensive account of their application in infectious disease epidemiology [6].

However, real-world outbreak surveillance systems rarely supply “clean” data in which all individuals share the same single-exposure event when estimating the incubation period. In practice, many datasets are inherently mixed: they contain not only secondary infections (Case 1), who were exposed at the original infection source, but also tertiary or later infections (Case 2), whose exposure stems from secondary transmissions. Data from Case 2 do not conform to the single-exposure assumption and thus act as outliers in the estimation process. Although excluding these outliers a priori would be ideal, the detailed investigations to confidently remove them are often expensive and time-consuming tasks, especially in urgent pandemic surveillance contexts. As a motivating example, Figure 1, which will be revisited in Section 4 of this paper, shows an epidemic curve of COVID-19 in Tianjin, China (21 January–12 February 2020), where four tertiary cases deviate considerably from the main cluster yet are not straightforward to exclude.

Our previous work tackled a similar situation using a three-parameter lognormal framework, proposing a robust estimation approach based on gamma-divergence, which is a robust divergence measure generalizing standard cross-entropy [8] that mitigated the impact of outliers on incubation period estimates [9]. While that approach proved valuable for distributions with long right tails, other scenarios may call for alternative parametric assumptions. In particular, the three-parameter Weibull distribution may be more suitable if the data lack extremely heavy tails or if epidemiological insight suggests that the hazard rate of onset changes monotonically over time rather than following the shape implied by the lognormal form. Nonetheless, just as with the lognormal model, conventional maximum likelihood methods applied to the Weibull distribution remain susceptible to contamination from tertiary infections. In addition, a general γ-divergence-based estimation framework, which can in principle be applied to various parametric models, has been proposed by Okuno (2023) [10]. Although this approach can be extended to outbreak datasets, it does not specifically target the optimization challenges posed by a three-parameter Weibull model under severe data contamination (e.g., tertiary infections). In this study, we focus on a dedicated method tailored to the three-parameter Weibull distribution and its estimation procedure. We develop a specialized majorization–minimization (MM) algorithm that guarantees a monotonic decrease in our γ-based objective function and also derive its covariance matrix. This specialization allows us to handle mixed or outlier-contaminated outbreak data more stably in practice while still benefiting from the robust properties of γ-divergence.

To address this issue, the current study adapts a γ-divergence-based robust methodology to the three-parameter Weibull setting, interpreting its location parameter as the unknown exposure time. We also develop a tailored MM algorithm to optimize a γ-cross-entropy criterion, enabling a stable estimation of the shape, scale, and location parameters—even under mixed or contaminated data conditions. Simulation experiments demonstrate that our approach substantially reduces bias and variance compared to traditional estimators, while an application to real-world surveillance data for COVID-19 further highlights its practicality.

This article is organized as follows: In Section 2, we review γ-divergence and introduce the associated objective function tailored to the three-parameter Weibull distribution of our interest. We then present an optimization method based on the MM algorithm. Next, a sandwich-type estimator for the covariance matrix is proposed using the theory of M-estimation. Simulation and real-world data analyses employing epidemiological surveillance data for COVID-19 and hepatitis A are described in Section 3 and Section 4, respectively. The article ends with a discussion in Section 5.

## 2. Method

### 2.1. Three-Parameter Weibull Distribution for Estimating the Exposure Time to Infectious Source and Incubation Period

Let $y_{i}$ be the disease onset timing of the *i*th individual (i=1,⋯,N) and assumed to have the probability density function (PDF) given by(1)$$\begin{array}{cc} f(y_{i}|\theta ) & =\alpha \beta {\left(y_{i}-\eta \right)}^{\beta -1}\exp\left\{-\alpha {\left(y_{i}-\eta \right)}^{\beta }\right\}\\ \; & \;(-\infty <\eta <y_{i}<\infty ,\alpha >0,\beta >0). \end{array}$$
The cumulative distribution function (cdf) and hazard function are, respectively,(2)$$\begin{array}{c} F\left(y\right)=1-\exp\left\{-\alpha {\left(y-\eta \right)}^{\beta }\right\} \end{array}$$
and(3)$$\begin{array}{c} h\left(y\right)=\alpha \beta {\left(y-\eta \right)}^{\beta -1}. \end{array}$$
The three-parameter Weibull distribution reduces to the two-parameter (i.e., conventional) Weibull distribution when η=0. Figure 2 illustrates the PDFs in Equation (Equation 1) when η=−1.

Here, we assume the single and simultaneous exposure to the infectious source: i.e., every individual was supposed to be exposed to one infectious source at the same time. Under this assumption of single-point exposure, the notable advantage of the three-parameter Weibull distribution is the fact that the support of *Y* (the disease onset timing) ranges [η,∞); therefore, η can be interpreted as “the timing of exposure to the infectious source”. Once the parameters are estimated, the average and q-percentile of the incubation period can be calculated, respectively, as(4)$$\begin{array}{c} {\left(\frac{1}{\alpha }\right)}^{1/\beta }\Gamma \left(1+\frac{1}{\beta }\right)+\eta \;\;\mathrm{and}\;{\left\{\frac{-\log(1-q)}{\alpha }\right\}}^{1/\beta }+\eta \end{array}$$
where *q* is the 100*q*% percentile of the three-parameter Weibull distribution.

### 2.2. Brief Introduction of γ-Divergence

The γ-divergence was defined for two PDFs by Fujisawa and Eguchi (2008) [11]. Let g(x) and f(x|θ) be the PDFs of the data-generating and parametric model distributions of *x*, respectively. The γ-divergence is defined by$$\begin{array}{cc} D_{\gamma }\left(g,f_{\theta }\right) & =\frac{1}{\gamma (1+\gamma )}\log\int g{\left(x\right)}^{1+\gamma }dx-\frac{1}{\gamma }\log\int g\left(x\right)f{\left(x|\theta \right)}^{\gamma }dx\\ \; & \;+\frac{1}{1+\gamma }\log\int f{\left(x|\theta \right)}^{1+\gamma }dx, \end{array}$$
where $f_{\theta }$ is the parametric PDF of interest. Note that $\lim_{\gamma \to 0}D_{\gamma }\left(g,f_{\theta }\right)=\int g\left(x\right)\log\frac{g(x)}{f(x|\theta )}dx$, which is the Kullback–Leibler (KL) divergence. This divergence satisfies the following two properties: (i) $D_{\gamma }\left(g,f_{\theta }\right)\geq D_{\gamma }\left(g,g\right)$ and (ii) $D_{\gamma }\left(g,f_{\theta }\right)=0\Leftrightarrow g\left(x\right)=cf\left(x|\theta \right)$, where *c* is a positive constant. Lower values of γ approach traditional likelihood-based methods such as KL divergence, which are efficient but less robust to outliers, while higher values prioritize robustness at the cost of efficiency.

Based on the above γ-divergence, the empirical version of γ-cross entropy between g(x) and f(x|θ) is defined as$$\begin{array}{c} d_{\gamma }\left(\bar{g},f_{\theta }\right)=-\frac{1}{\gamma }\log\sum_{i=1}^{N}f{\left(x_{i}|\theta \right)}^{\gamma }+\frac{1}{1+\gamma }\log\int_{X}f{\left(x|\theta \right)}^{1+\gamma }dx, \end{array}$$
where $\bar{g}$ is the empirical PDF. The robust estimator (γ-estimator) is then defined as(5)$$\begin{array}{c} {\hat{\theta }}_{\gamma }=\underset{\theta }{\mathrm{argmin}}d_{\gamma }\left(\bar{g},f_{\theta }\right). \end{array}$$

Now, we consider the case where the data-generating distribution is contaminated with outliers (i.e., Case 2) and given by(6)$$\begin{array}{c} g\left(x\right)=\left(1-\varepsilon \right)f_{\theta }\left(x\right)+\varepsilon \delta \left(x\right), \end{array}$$
which is a mixture of the target distribution $f_{\theta }\left(x\right)$ and certain contamination distribution δ(x), and ε denotes the proportion of outliers. The most important assumption here is(7)$$\begin{array}{c} \nu_{f_{\theta }}={\left\{\int \delta \left(x\right)f_{\theta }{\left(x\right)}^{\gamma }dx\right\}}^{1/\gamma }\approx 0\;\;\mathrm{for}\;\mathrm{an}\;\mathrm{appropriately}\;\mathrm{large}\;\gamma , \end{array}$$
which assumes the practical situation where the outliers mostly lie on the tail of the target distribution. Kanamori and Fujisawa (2015) show the robust properties from a viewpoint of latent bias [12].

### 2.3. γ-Entropy and MM Algorithm for Optimization

Using (Equation 1), the γ-cross entropy function can be written as(8)$$\begin{array}{cc} l_{\gamma }\left(\theta \right) & =-\frac{1}{\gamma }\log\left(\frac{1}{n}\sum_{i=1}^{N}f{\left(y_{i}|\theta \right)}^{\gamma }\right)+\frac{1}{1+\gamma }\log\left(\int_{\eta }^{\infty }f{\left(y|\theta \right)}^{1+\gamma }dy\right)\\ \; & =-\frac{1}{\gamma }\log\left(\frac{1}{n}\sum_{i=1}^{N}f{\left(y_{i}|\theta \right)}^{\gamma }\right)+\frac{1}{1+\gamma }\log\left[\alpha^{\frac{\gamma }{\beta }}\beta^{\gamma }{\left(1+\gamma \right)}^{-(1+\gamma -\frac{\gamma }{\beta })}\Gamma \left(1+\gamma -\frac{\gamma }{\beta }\right)\right] \end{array}$$
Notably, the second term in Equation (Equation 8) can be written in the simple form here, while it is not possible in many cases. To obtain the minimizer, we propose the iterative algorithm of the majorization–minimization algorithm (MM algorithm) as follows.

Let us prepare the majorization function *h* for the cross-entropy satisfying$$\begin{array}{c} h\left(\theta^{\left(t\right)}|\theta^{\left(t\right)}\right)=l_{\gamma }\left(\theta^{\left(t\right)}\right)\\ h\left(\theta |\theta^{\left(t\right)}\right)\geq l_{\gamma }\left(\theta \right), \end{array}$$
where $\theta^{\left(t\right)}$ is the parameter value of the *t*-th iteration step for t=0,1,2,⋯. The MM algorithm applies the iterative procedure by$$\begin{array}{c} \theta^{\left(t+1\right)}=\underset{\theta }{\mathrm{argmin}}h\left(\theta |\theta^{\left(t\right)}\right). \end{array}$$
It is possible to show that the objective function $l_{\gamma }\left(\theta \right)$ monotonically decreases at each step because$$\begin{array}{c} l_{\gamma }\left(\theta^{\left(t\right)}\right)=h\left(\theta^{\left(t\right)}|\theta^{\left(t\right)}\right)\geq h\left(\theta^{\left(t+1\right)}|\theta^{\left(t\right)}\right)\geq l_{\gamma }\left(\theta^{\left(t+1\right)}\right). \end{array}$$

Here, we propose the majorization function for Equation (Equation 8) using Jensen’s inequality as follows: $$\begin{array}{cc} l_{\gamma }\left(\theta \right) & \leq -\frac{1}{n\gamma }\sum_{i=1}^{N}w_{i}\log\left(f{\left(y_{i}|\theta \right)}^{\gamma }\frac{\sum_{j=1}^{N}f{\left(y_{j}|\theta^{\left(t\right)}\right)}^{\gamma }}{f{\left(y_{i}|\theta^{\left(t\right)}\right)}^{\gamma }}\right)+\\ \; & \;\frac{1}{1+\gamma }\log\left[\alpha^{\frac{\gamma }{\beta }}\beta^{\gamma }{\left(1+\gamma \right)}^{-(1+\gamma -\frac{\gamma }{\beta })}\Gamma \left(1+\gamma -\frac{\gamma }{\beta }\right)\right]\\ \; & =-\sum_{i=1}^{N}w_{i}\log f(y_{i}|\theta )+\frac{1}{1+\gamma }\log\left[\alpha^{\frac{\gamma }{\beta }}\beta^{\gamma }{\left(1+\gamma \right)}^{-(1+\gamma -\frac{\gamma }{\beta })}\Gamma \left(1+\gamma -\frac{\gamma }{\beta }\right)\right]+c\left(\theta^{\left(t\right)}\right)\\ \; & =h\left(\theta |\theta^{\left(t\right)}\right)+c\left(\theta^{\left(t\right)}\right), \end{array}$$
where $w_{i}=\frac{f{\left(y_{i}|\theta^{\left(t\right)}\right)}^{\gamma }}{\sum_{j=1}^{N}f{\left(y_{j}|\theta^{\left(t\right)}\right)}^{\gamma }}$ and $c(\theta^{\left(t\right)})$ is a term that does not depend on the parameter θ. The first term on the target function $l_{\gamma }\left(\theta \right)$ is a mixture of densities, which is not easy to be optimized in general, while the first term in $h(\theta |\theta^{\left(t\right)})$ is a weighted log-likelihood and is easy to be optimized by using the derivatives in the Appendix A.

Using the *t*-th iteration values of $\alpha^{\left(t\right)}$, the t+1-th iteration values can be obtained as follows:(9)$$\begin{array}{cc} \; & \alpha^{\left(t+1\right)}=\frac{1-\frac{\gamma }{\beta^{\left(t\right)}\left(1+\gamma \right)}}{\sum_{i=1}^{N}w_{i}{\left(y_{i}-\eta^{\left(t\right)}\right)}^{\beta^{\left(t\right)}}} \end{array}$$

Additionally, at the t+1-th iteration, the values of $\beta^{\left(t+1\right)}$ and $\eta^{\left(t+1\right)}$ can be obtained by finding the root of the following equations: (10)$$\begin{array}{cc} \; & -\sum_{i=1}^{N}w_{i}\left\{\frac{1}{\beta^{\left(t+1\right)}}+\log\left(y_{i}-\eta^{\left(t\right)}\right)-\alpha^{\left(t\right)}{\left(y_{i}-\eta^{\left(t\right)}\right)}^{\beta^{\left(t+1\right)}}\log\left(y_{i}-\eta^{\left(t\right)}\right)\right\}+\\ \; & \;\frac{1}{1+\gamma }\{-\frac{\gamma }{\beta^{2(t+1)}}\log\alpha^{\left(t\right)}+\frac{\gamma }{\beta^{\left(t+1\right)}}-\frac{\gamma }{\beta^{2(t+1)}}\log\left(1+\gamma \right)+\\ \; & \;\frac{\gamma }{\beta^{2(t+1)}}\psi \left(1+\frac{\gamma (\beta^{\left(t+1\right)}-1)}{\beta^{\left(t+1\right)}}\right)\}=0 \end{array}$$(11)$$\begin{array}{cc} \; & \sum_{i=1}^{N}w_{i}\left\{\frac{\beta^{\left(t\right)}-1}{y_{i}-\eta^{\left(t+1\right)}}-\alpha^{\left(t\right)}\beta^{\left(t\right)}{\left(y_{i}-\eta^{\left(t+1\right)}\right)}^{\beta^{\left(t\right)}-1}\right\}=0. \end{array}$$

**Remark 1** (Identification of outliers). *By plugging the estimated ${\hat{\theta }}_{\gamma }$ into Equation (1) (or the associated likelihood function), we can visually inspect the presence or absence of outliers. If a sample is an outlier, it will be plotted at the tail of the distribution. By using the estimated ε, we can identify 100(1−ε)% cases as the smallest estimated values of $f(y_{i}|{\hat{\theta }}_{\gamma })$ as outliers.*

**Remark 2** (Selection criterion for tuning parameter γ). *From the result of Sugasawa and Yonekura (2021) [13], we use the following selection criterion:*(12)$$\begin{array}{cc} H_{N}\left(\gamma \right) & =\sum_{i=1}^{N}[f^{\prime }{\left(y_{i}|{\hat{\theta }}_{\gamma }\right)}^{2}f{\left(y_{i}|{\hat{\theta }}_{\gamma }\right)}^{\gamma -2}\left\{\frac{2(\gamma -1)}{C_{\gamma }\left({\hat{\theta }}_{\gamma }\right)}+\frac{f{\left(y_{i}|{\hat{\theta }}_{\gamma }\right)}^{\gamma }}{C_{\gamma }\left({\hat{\theta }}_{\gamma }\right)}\right\}\\ \; & +\frac{2f{\left(y_{i}|{\hat{\theta }}_{\gamma }\right)}^{\gamma -1}f^{\prime \prime }\left(y_{i}|{\hat{\theta }}_{\gamma }\right)}{C_{\gamma }\left({\hat{\theta }}_{\gamma }\right)}], \end{array}$$*where*
$$\begin{array}{cc} \; & C_{\gamma }\left({\hat{\theta }}_{\gamma }\right)={\left\{\int_{X}f{\left(x|{\hat{\theta }}_{\gamma }\right)}^{1+\gamma }dx\right\}}^{\frac{\gamma }{1+\gamma }}={\left\{{\hat{\alpha }}^{\frac{\gamma }{\hat{\beta }}}{\hat{\beta }}^{\gamma }{\left(1+\gamma \right)}^{-(1+\gamma -\frac{\gamma }{\hat{\beta }})}\Gamma \left(1+\gamma -\frac{\gamma }{\hat{\beta }}\right)\right\}}^{\frac{\gamma }{1+\gamma }},\\ \; & f^{\prime }\left(y_{i}|{\hat{\theta }}_{\gamma }\right)=\left(\frac{\hat{\beta }}{y_{i}-\hat{\eta }}-\hat{\alpha }\hat{\beta }\right)f\left(y_{i}|{\hat{\theta }}_{\gamma }\right),\;\;and\;\;f^{\prime \prime }\left(y_{i}|{\hat{\theta }}_{\gamma }\right)={\left(\frac{\hat{\beta }}{y_{i}-\hat{\eta }}-\hat{\alpha }\hat{\beta }\right)}^{2}f\left(y_{i}|{\hat{\theta }}_{\gamma }\right). \end{array}$$
*Then, we estimate the optimal γ as $\gamma_{opt}={\mathrm{argmin}}_{\gamma }H_{N}\left(\gamma \right)$. In practice, as in the simulation section, several γ values are prepared a priori, and the one with the smallest $H_{N}\left(\gamma \right)$ is selected. Other simple decision rules might be based on previous studies using gamma divergence [14] and density power divergence [15,16], in which the value of 0.5 was selected as a reasonable middle ground for practical applications, offering sufficient robustness without a significant loss of efficiency. In addition, other practical approaches for tuning γ, such as cross-validation or utilizing external validation datasets, are also viable [14].*
*Note that $H_{N}\left(\gamma \right)$ is known as the Hyvarinen score, which is a model selection criterion defined via the score function (from a Bayesian perspective). Further, note also that while γ-divergence is known to exhibit relative affine invariance [11], the selection criterion $H_{N}\left(\gamma \right)$ itself does not necessarily share this property; thus, rescaling data (e.g., doubling the observed time) may alter the optimal value of γ.*


### 2.4. Initial Value of MM Algorithm

The proposed MM algorithm ensures the monotonic decreasing property of the objective function. However, when Equation (Equation 8) has several local minima, it converges to a local minimum rather than the global minimum. Hence, the selection of the initial value is essential. A simple approach is to start from the estimated values using the maximum likelihood method or method of moments, which is applied in the simulation section. Another and more complex approach is to run the MM algorithm with various initial values and select the best run with the smallest value of $l_{\gamma }$. The procedure for creating the initial values is as follows. First, a subsample is created by randomly selecting *q* samples from *N* observations. Next, the median of each subsample is used as the initial value for $\alpha^{\left(0\right)}$, the median absolute deviation as the initial value for $\beta^{\left(0\right)}$, and $\min\left(y_{i}\right)-\kappa \;\left(\kappa >0\right)$ as the initial value for $\eta^{\left(0\right)}$, and then we calculate the minimum value of Equation (Equation 8). The value of κ should be determined beforehand based on an expert opinion or other criteria. Repeat the above process *M* times and select the initial values that yield the smallest value of Equation (Equation 8).

### 2.5. Asymptotic Properties of ${\hat{\theta }}_{\gamma }$

We consider the estimation of the covariance matrix of ${\hat{\theta }}_{\gamma }$. Let us assume some regularity conditions, which are common in the M-estimator (more precisely, in the theory of normalized estimating equation [17]). The asymptotic normality of ${\hat{\theta }}_{\gamma }$ is given as $\sqrt{N}\left({\hat{\theta }}_{\gamma }-\theta_{\gamma }^{*}\right)\overset{d}{\to }N\left(0,\Sigma_{\gamma }\right)$, where $\Sigma_{\gamma }=G_{\gamma }^{-1}U_{\gamma }G_{\gamma }^{-1}$, $G_{\gamma }=E\left[\frac{\partial^{2}l_{\gamma }\left(\theta \right)}{\partial \theta \partial \theta^{T}}\right]$ and $U_{\gamma }=E\left[\frac{\partial l_{\gamma }\left(\theta \right)}{\partial \theta }\frac{\partial l_{\gamma }\left(\theta \right)}{\partial \theta^{T}}\right]$. The asymptotic covariance matrix, $\Sigma_{\gamma }$, can then be estimated using the sandwich-type estimator:(13)$$\begin{array}{c} {\hat{\Sigma }}_{\gamma }={\left\{{\hat{G}}_{\gamma }\left({\hat{\theta }}_{\gamma }\right)\right\}}^{-1}{\hat{U}}_{\gamma }\left({\hat{\theta }}_{\gamma }\right){\left\{{\hat{G}}_{\gamma }\left({\hat{\theta }}_{\gamma }\right)\right\}}^{-1}, \end{array}$$
where ${\hat{H}}_{\gamma }\left({\hat{\theta }}_{\gamma }\right)$ and ${\hat{G}}_{\gamma }\left({\hat{\theta }}_{\gamma }\right)$ can be empirically estimable. The detailed derivations are in the Appendix B.

## 3. Monte Carlo Simulation Experiments

### 3.1. Simulation Setup

To assess the performance of our approach, we conducted Monte Carlo simulation experiments, varying three key parameters: the proportion of outliers, ε∈{0.05,0.1,0.3}; the value of η in the distribution that generates the outliers (thus it is denoted as $\eta_{\mathrm{out}}$); and the number of non-outlier samples, N∈{20,50,200}. In particular, the main body of the data—denoted by $f_{\theta }\left(x\right)$ in Equation (Equation 6)—is assumed to follow a three-parameter Weibull distribution with parameters α=0.111, σ=3, and η=−1. Conversely, the outliers are drawn from the same three-parameter Weibull distribution except that $\eta_{\mathrm{out}}$ takes values from the set {0,1,3}. Note that in principle, the distribution of outliers could be any form as long as it appears in the tail of the main distribution; thus, the Weibull assumption for the outliers is adopted here primarily for simplicity rather than necessity. Moreover, since the same pathogen is implicated in both secondary (Case 1) and tertiary or subsequent (Case 2) infections, applying the same distribution is justified. Overall, considering all combinations of N,ε, and $\eta_{\mathrm{out}}$ yields a total of 27 scenarios (see Table 1). For each scenario, 1000 Monte Carlo simulations were performed.

For each scenario, the procedure to generate an individual dataset for the *k*th scenario, $y^{\left(k\right)}=\left(y_{1}^{\left(k\right)},\cdots ,y_{N}^{\left(k\right)}\right)$, is as follows. First, we randomly generate *N* samples from a three-parameter Weibull distribution with parameters (α=0.111,σ=3,η=−1). Next, we randomly generate Nε samples from a three-parameter Weibull distribution with parameters $\left(\alpha =0.111,\sigma =3,\eta^{\left(k\right)}\in \left\{0,1,3\right\}\right)$ corresponding to the *k*th scenario.

We evaluated the performance in terms of the bias and mean squared error (MSE) of estimated mean and 95% percentile values of the true distribution, which were calculated from Equation (Equation 4) and $\hat{\eta }$. The comparison methods included estimates based on (1) *ml*: maximum likelihood method [18,19], (2) *mm*: method of moments [20], and (3) *mps*: the method of maximum product spacing [21,22]. These conventional methods are easily implemented in the *R* program [23] using the fitWeibull() function in the ForestFit package [24].

### 3.2. Simulation Results

The Monte Carlo simulation results show that the proposed method consistently outperforms the conventional approaches across most scenarios, providing less biased and more efficient estimates for both mean and 95% percentile of the true distribution (defined in Equation (Equation 6)) and $\hat{\eta }$. As shown in Table 2 and Table 3, our method achieves the smallest bias and MSE on average. Specifically, the bias in the mean under our approach achieved 65% reduction compared with that of the conventional approaches; the bias for our approach ranges from 0.002 to 0.363 (mean 0.106) compared to ranges of 0.036 to 0.937 (mean 0.288) for *ml*, 0.036 to 0.926 (mean 0.285) for *moment*, and 0.047 to 1.182 (mean 0.361) for *mps*. The difference in performance between our approach and conventional approaches is even more pronounced in the estimation of the 95% percentile, achieving a 68% reduction compared with that of the conventional approaches; the bias for our approach ranges from 0.010 to 0.960 (mean 0.268) compared to ranges of 0.077 to 3.346 (mean 0.979) for *ml*, −0.340 to 1.903 (mean 0.316) for *moment*, and 0.137 to 4.365 (mean 1.238) for *mps*. Regarding the bias of $\hat{\eta }$, our approach also outperforms the conventional approaches, achieving nearly a 90% reduction (excluding the results of *mps*, which show extremely large biases): the bias for our approach ranges from −0.080 to 0.321 (mean 0.086) compared to ranges of −0.445 to 0.586 (mean 0.164) for *ml*, 1.458 to 1.734 (mean 1.550) for *moment*, and −4509.500 to 1.017 (mean −2532.864) for *mps*.

In terms of MSE, our method also provides markedly smaller values in the estimation of the mean (a 70% reduction overall), ranging from 0.002 to 0.407 (mean 0.054) compared to ranges of 0.004 to 0.884 (mean 0.162) for *ml*, 0.004 to 0.874 (mean 0.159) for *moment*, and 0.005 to 1.422 (mean 0.230) for *mps*. Again, the performance gap is even enlarged for the 95% percentile, achieving a 76% reduction in MSE compared with the conventional approaches; MSE of our approach range from 0.013 to 3.175 (mean 0.418) compared to ranges of 0.019 to 11.315 (mean 1.882) for *ml*, 0.004 to 3.664 (mean 0.560) for *moment*, and 0.027 to 19.408 (mean 2.806) for *mps*. Regarding the MSE of $\hat{\eta }$, our approach also outperforms the conventional approaches, achieving an 88% reduction: the MSE for our approach ranges from 0.035 to 0.496 (mean 0.194), compared to ranges of 0.032 to 9.238 (mean 0.843) for *ml*, 2.134 to 3.008 (mean 2.421) for *moment*, and 0.401 to $10^{12}$ (mean $10^{13}$) for *mps*.

Overall, conventional methods tend to suffer in performance under small sample sizes, when the proportion of outliers is high, or when outliers are not concentrated in the tail of the target distribution (i.e., small η). In contrast, our method remains robust and yields stable estimates even under these challenging conditions. In addition, although all methods show improved performance (i.e., reduced bias and MSE) as *N* increases, our method continues to provide similar or superior performance compared to the conventional methods even at N=200 and smaller outlier proportions ε=0.05 or 0.1 (Scenarios 3 and 6).

## 4. Application for Real-World Data: Epidemiological Surveys for COVID-19

This section applies both the proposed method and the comparison methods to contact tracing surveillance for COVID-19, aiming to identify infection sources and estimate the incubation period. Contact tracing surveillance refers to investigations primarily conducted by local public health authorities to prevent disease spread in communities where infections have occurred. The data and corresponding R code (https://www.r-project.org/, accessed on 17 February 2025) are available on the corresponding author’s GitHub page (https://github.com/, accessed on 17 February 2025).

We focus on a COVID-19 outbreak in Tianjin Province, mainland China, from 21 January to 12 February 2020, as detailed by Wang and Teunis (2020) [7]. The dataset, consisting of 112 confirmed cases, highlights a mixture of secondary, tertiary, and subsequent infection cases with the transmission network inferred from symptom onset dates. Figure 1 displays the epidemic curve: the first exposure is designated as day 0, and the reported onset dates span day 1 to day 24. In the figure, 31 secondary infection cases are shown in gray, whereas 39 tertiary or subsequent infection cases are shown in black.

To estimate the exact exposure time, the tertiary and subsequent cases would ideally be excluded, but doing so requires extensive investigation, which is typically time consuming and expensive in the context of emerging infectious diseases. Consequently, using a “mixed” dataset without removing these cases is common in practice as is treating the tertiary and subsequent cases as outliers in the analysis.

Table 4 briefly summarizes a comparison of the estimated time of exposure to the infectious sources, $\hat{\eta }$, across our approach and the conventional methods. With the selected γ of 1, our method produces the estimates of η with the corresponding 95% CI for η, mean and 95% percentile of the distribution of incubation period: ${\hat{\eta }}^{our}=-0.19\;\left(95\%\mathrm{CI}:-3.14,2.76\right)$, and the mean and 95% percentile are 3.96 and 9.77. We note that the estimated ${\hat{\eta }}^{our}$ is quite close to the actual time of exposure (day 0). In contrast, the conventional approaches produce MLE values of η ranging from −0.49 to 2.20. Clearly, our method succeeds in returning the preferable estimated value of η closer to the realistic exposure time. In terms of the distribution of the incubation period, our method provides a mean and 95% percentile of 3.96 and 9.77, respectively, whereas the conventional methods provide estimates ranging from 3.89 to 4.28 for the mean and from 6.96 to 10.63 for the 95% percentile.

Our robust estimation method has practical implications for understanding the biological evolution of COVID-19, particularly regarding changes in incubation period distributions associated with different SARS-CoV-2 variants. Accurately estimating exposure times and incubation periods in contaminated datasets enables epidemiologists to better capture subtle shifts in viral characteristics, such as transmissibility and generation intervals, across successive infection generations. Such insights can be crucial when assessing how viral mutations or emerging variants alter epidemiological parameters, influencing the trajectory of outbreaks and informing timely public health responses.

## 5. Discussion

We have introduced a novel robust approach for estimating both the exposure time to infectious sources and the incubation period based on the γ-divergence approach for the Weibull distribution. This approach maintains robustness even under substantial contamination, which often arises when unexpected secondary or tertiary cases are captured in rapid epidemiological surveillance. A frequent challenge in robust estimation lies in developing an efficient algorithm, especially given the non-convex and non-differentiable nature of many robust objective functions. In this study, we devised a practical estimation method that guarantees a monotonic decrease in the objective function by leveraging the MM algorithm.

Although our analysis assumed a contaminated density of the form $g\left(x\right)=\left(1-\varepsilon \right)f_{\theta }\left(x\right)+\varepsilon \delta \left(x\right)$, we note that this setup can be generalized to a more intricate mixture, $g\left(x\right)=\left(1-\sum_{j=1}^{k}\varepsilon_{j}\right)f_{\theta }\left(x\right)+\sum_{j=1}^{k}\varepsilon_{j}\delta_{j}\left(x\right)$, using essentially the same framework. Numerical simulations and applications to real-world data consistently indicate that our method surpasses conventional estimators in terms of bias, MSE, and both the mean and 95% percentile of the true distribution.

A key remaining question in the realm of infectious disease surveillance involves the practical selection of the incubation period distribution. Currently, we assume that the incubation period follows the Weibull distribution. However, the accuracy of the estimation strongly depends on the validity of this assumed distribution. If the true distribution deviates from the assumed one, the estimates of exposure time may be biased or imprecise, resulting in an inaccurate estimation of the incubation period. Our robust approach can be extended to other types of distributions. Although the idea is straightforward, further efforts are required to derive a new MM algorithm and covariance matrices. This will be the subject of our future research.

## Figures and Tables

**Figure 1 entropy-27-00321-f001:**
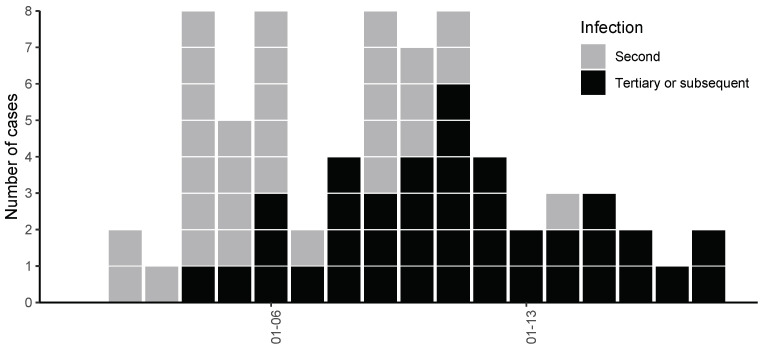
Epicurve of COVID-19 in China [7].

**Figure 2 entropy-27-00321-f002:**
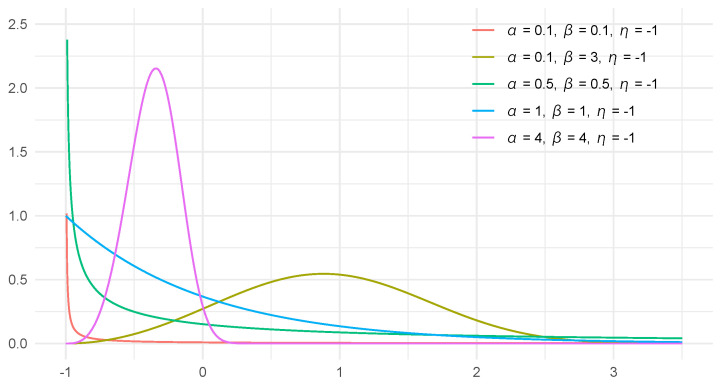
Probability density functions of three-parameter Weibull distribution when η=−1.

**Table 1 entropy-27-00321-t001:** Settings in 27 Monte Carlo simulation scenarios.

Scenario	Samples	Parameter	
	N	ε	$\eta^{\left(k\right)}$
1	20	0.05	0
2	50	0.05	0
3	200	0.05	0
4	20	0.1	0
5	50	0.1	0
6	200	0.1	0
7	20	0.3	0
8	50	0.3	0
9	200	0.3	0
10	20	0.05	1
11	50	0.05	1
12	200	0.05	1
13	20	0.1	1
14	50	0.1	1
15	200	0.1	1
16	20	0.3	1
17	50	0.3	1
18	200	0.3	1
19	20	0.05	3
20	50	0.05	3
21	200	0.05	3
22	20	0.1	3
23	50	0.1	3
24	200	0.1	3
25	20	0.3	3
26	50	0.3	3
27	200	0.3	3

**Table 2 entropy-27-00321-t002:** Results of 27 Monte Carlo scenarios: bias.

Scenario	Mean (True = 0.858)				95% Percentile (True = 2.000)				η (True = −1)			
	Our	ml	Moment	mps	Our	ml	Moment	mps	Our	ml	Moment	mps
1	0.039	0.046	0.047	0.122	0.064	0.089	−0.327	0.346	−0.080	−0.445	1.467	−11,091.270
2	0.035	0.036	0.036	0.074	0.080	0.077	−0.340	0.190	−0.013	−0.070	1.458	−0.119
3	0.042	0.045	0.046	0.047	0.095	0.107	−0.320	0.137	0.025	0.008	1.464	0.009
4	0.074	0.084	0.085	0.160	0.144	0.181	−0.259	0.443	0.000	−0.158	1.491	−44,509.500
5	0.086	0.091	0.090	0.128	0.187	0.196	−0.248	0.314	0.050	0.017	1.495	0.217
6	0.088	0.094	0.094	0.096	0.187	0.207	−0.241	0.239	0.056	0.029	1.498	0.051
7	0.223	0.231	0.231	0.297	0.421	0.441	−0.042	0.675	0.011	−0.196	1.597	−5137.573
8	0.223	0.227	0.227	0.261	0.451	0.453	−0.037	0.566	0.073	0.053	1.590	0.260
9	0.226	0.231	0.231	0.234	0.450	0.458	−0.033	0.490	0.080	0.042	1.595	0.056
10	0.042	0.086	0.085	0.200	0.129	0.331	−0.167	0.698	0.090	0.087	1.470	−2361.782
11	0.046	0.080	0.081	0.145	0.146	0.301	−0.187	0.475	0.137	0.120	1.475	−4377.724
12	0.047	0.094	0.094	0.099	0.156	0.366	−0.142	0.420	0.132	0.104	1.479	0.198
13	0.098	0.181	0.180	0.314	0.287	0.623	0.045	1.053	0.161	0.252	1.511	−919.006
14	0.105	0.188	0.187	0.266	0.325	0.638	0.059	0.851	0.220	0.227	1.521	0.664
15	0.092	0.183	0.182	0.188	0.281	0.628	0.053	0.683	0.191	0.146	1.519	0.229
16	0.363	0.467	0.465	0.587	0.960	1.291	0.537	1.722	0.297	0.355	1.673	0.745
17	0.334	0.465	0.462	0.534	0.926	1.282	0.539	1.500	0.321	0.282	1.672	0.643
18	0.243	0.464	0.461	0.469	0.672	1.268	0.539	1.315	0.264	0.185	1.670	0.222
19	0.022	0.193	0.189	0.387	0.051	0.976	0.346	1.595	0.027	0.448	1.482	0.974
20	0.008	0.161	0.157	0.266	0.014	0.863	0.252	1.124	0.006	0.314	1.491	0.809
21	0.002	0.196	0.190	0.206	0.011	1.012	0.361	1.088	0.005	0.214	1.497	0.311
22	0.042	0.374	0.367	0.605	0.129	1.637	0.832	2.453	0.015	0.528	1.510	1.017
23	0.011	0.377	0.367	0.498	0.020	1.635	0.830	1.988	−0.009	0.373	1.521	0.836
24	0.003	0.376	0.363	0.386	0.016	1.624	0.824	1.704	0.023	0.250	1.524	0.320
25	0.329	0.931	0.926	1.182	0.894	3.346	1.903	4.365	0.164	0.586	1.716	1.002
26	0.047	0.929	0.920	1.052	0.118	3.236	1.879	3.724	0.054	0.414	1.727	0.779
27	0.003	0.937	0.923	0.946	0.010	3.177	1.880	3.258	0.018	0.270	1.734	0.304

**Table 3 entropy-27-00321-t003:** Results of 27 Monte Carlo scenarios: MSE.

Scenario	Mean (True = 0.858)				95% Percentile (True = 2.000)				η (True = −1)			
	Our	ml	Moment	mps	Our	ml	Moment	mps	Our	ml	Moment	mps
1	0.03	0.02	0.02	0.04	0.13	0.08	0.15	0.25	0.496	9.238	2.175	6,626,303,000.000
2	0.01	0.01	0.01	0.02	0.06	0.04	0.13	0.08	0.248	0.385	2.134	55.854
3	0.00	0.00	0.00	0.00	0.02	0.02	0.11	0.03	0.037	0.038	2.145	0.049
4	0.03	0.03	0.03	0.05	0.15	0.11	0.11	0.34	0.439	2.822	2.247	1,051,845,000,000.000
5	0.02	0.02	0.02	0.03	0.08	0.07	0.08	0.14	0.191	0.281	2.246	0.858
6	0.01	0.01	0.01	0.01	0.04	0.05	0.06	0.07	0.038	0.038	2.245	0.050
7	0.07	0.07	0.07	0.11	0.30	0.27	0.04	0.57	0.461	4.174	2.569	4,799,133,000.000
8	0.06	0.06	0.06	0.08	0.26	0.23	0.02	0.36	0.155	0.183	2.536	0.428
9	0.05	0.06	0.06	0.06	0.21	0.22	0.00	0.25	0.035	0.032	2.545	0.041
10	0.03	0.03	0.03	0.06	0.18	0.20	0.07	0.67	0.369	2.160	2.185	3,927,284,000.000
11	0.01	0.02	0.02	0.03	0.08	0.12	0.05	0.27	0.126	0.172	2.186	19,169,410,000.000
12	0.01	0.01	0.01	0.01	0.04	0.14	0.02	0.19	0.042	0.039	2.188	0.072
13	0.05	0.05	0.05	0.12	0.32	0.48	0.05	1.29	0.276	0.464	2.306	845,950,400.000
14	0.03	0.04	0.04	0.08	0.20	0.44	0.02	0.78	0.128	0.148	2.321	0.589
15	0.01	0.04	0.04	0.04	0.11	0.40	0.01	0.48	0.055	0.045	2.310	0.083
16	0.22	0.24	0.24	0.37	1.65	1.76	0.33	3.14	0.253	0.354	2.822	0.803
17	0.16	0.22	0.22	0.30	1.21	1.68	0.31	2.30	0.171	0.159	2.803	0.551
18	0.08	0.22	0.21	0.22	0.65	1.62	0.29	1.74	0.086	0.054	2.792	0.079
19	0.03	0.06	0.06	0.17	0.18	1.03	0.17	2.73	0.335	0.341	2.219	1.004
20	0.01	0.03	0.03	0.08	0.07	0.77	0.08	1.31	0.169	0.162	2.232	0.725
21	0.00	0.04	0.04	0.05	0.01	1.03	0.14	1.19	0.052	0.065	2.243	0.124
22	0.04	0.16	0.16	0.39	1.58	2.77	0.74	6.26	0.336	0.389	2.302	1.079
23	0.01	0.15	0.14	0.26	0.07	2.71	0.71	4.01	0.178	0.192	2.321	0.762
24	0.00	0.14	0.13	0.15	0.01	2.65	0.68	2.91	0.047	0.079	2.324	0.132
25	0.41	0.88	0.87	1.42	3.18	11.32	3.66	19.41	0.282	0.437	2.967	1.056
26	0.05	0.87	0.85	1.15	0.48	10.51	3.55	14.40	0.170	0.217	2.989	0.690
27	0.00	0.88	0.85	0.90	0.02	10.10	3.54	10.63	0.056	0.088	3.008	0.118

**Table 4 entropy-27-00321-t004:** Results of real-world data analysis of COVID-19.

		COVID-19
Susceptive exposure day		0
Sample size	N	70
	Secondary infection	31
	Tertiary and beyond infection	39
η	Our	−0.19
	ml	−0.49
	moment	2.20
	mps	0.70
Estimated incubation period: mean	Our	3.96
	ml	3.90
	moment	3.89
	mps	4.28
Estimated incubation period: 95% percentile	Our	9.77
	ml	8.95
	moment	6.96
	mps	10.63

## Data Availability

The data and corresponding R code are available on the corresponding author’s GitHub page (https://github.com/kingqwert/R/tree/master/Robust_3ParWeibull/, accessed on 17 February 2025).

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
