# Peer review of "Weibull-Type Incubation Period and Time of Exposure Using γ-Divergence"

_entropy, 2025, doi:10.3390/e27030321_

Round 1
Reviewer 1 Report
Comments and Suggestions for Authors
The paper describes a method to estimate incubation time and time of exposure by compensating for tertiary infections on Weibull-distributed data. Authors proposed to use gamma-divergence and devised a specific algorithm to estimate the parameters of the distribution based on such divergence. The paper is well written and the methodology mathematically sound. There are a few open points which require clarification. Most of these points relate to the assumptions of the proposed approach.
-Line 135. I suppose that theta_hat_gamma is the optimal value of theta found after MM with setting that specific value of gamma. Is this so? It was never introduced.
-Can you please briefly explain what H_n in eq. (12) actually quantifies? In addition, it seems the right-hand side of H_n does not depend on "n". Is "n" a variable or a name?
-Related to the previous question. Is gamma necessary in eq. (7)? The assumption that outliers mostly lie on the tail of the target distribution might not hold. Indeed, with a gamma sufficiently high (or low), you may get that integral close to 0. In addition, eq. (7) is monotonically increasing (or decreasing) with gamma. Does H_n have the same behavior? I expect not, otherwise it would be impossible to select the optimal gamma without a validation set. Please, clarify these points.
-As stated by the authors, it makes sense to have the outlier distribution as similar to the main distribution but shifted in time. Using this approach, one can simply use MLE method by using a mixture of Weibull distributions. Is this a feasible strategy? I am not asking to do any simulations but just an opinion (not necessary to modify the text if you wish so).
-I suppose H_gamma(theta_hat) in eq. (13) is not the same H_n. Is this so? Perhaps using different notation would help not getting confused.
-Please add the optimal value of gamma in the Tables.
-Line 174. It is said "the value of eta in the distribution that generates the outliers". It is delta that generates the outlier, isn't it? It is explained later that the delta distribution was defined as the same Weibull distribution but with eta changed. I was very confused when I read it. Perhaps revising the text by first saying how delta was defined and then defining the outlier parameter eta.
-The analysis on COVID data is very interesting. However, I have a few questions on this analysis. Please, reply to all point-by-point.
--As far as I understood, the data have been time aligned to have eta approximately 0 days. How this time alignment was performed? Have you used a prior information on the exposure time of COVID? Any particular reference?
--In a real scenario, one would like to estimate the time of exposure (which is a unknown information). So I guess, one would consider a maximum exposure window rather large (let's say 15/30 days). However, the three-parameter Weibull distribution is 0 for y<eta. So the estimate of eta is valid only if the COVID infection well fits the Weibull distribution (as correctly stated in the discussion). Are there any evidence of this in the COVID literature?
--What was the value of gamma found for COVID data? This would help reproducibility of your study.
--Does the value of gamma change if you time align the data differently? Let's say double of its used amount?
--I expect that eq. (7) would be a reasonable assumption only in certain situations. For example, if the exposure time is rather short and incubation time long, you may have many tertiary infections overlapped with the first batch of exposed subjects. Having this COVID data at disposal, is it possible to quantify such overlap? Is the assumption holding for COVID?
-Typos and others:
--Typically, journals require figures to be referenced in the order of appereance. Figure 1 has not been referenced before Figure 2.
--criteria is plural in Remark 2. I think you meant "criterion" unless H_n was defined as composition of criteria.
--Table 4. esultsofreal-world
--I suggest to use the actual Greek letters in Fig. 1.
Reviewer 2 Report
Comments and Suggestions for Authors
The authors of this article introduced a robust estimation framework based on a three-parameter Weibull distribution in which the location parameter naturally corresponds to the unknown exposure time. The method employs a γ-divergence-based objective function with a majorization-minimization (MM) algorithm designed to guarantee a monotonic decrease in the objective function despite the nonconvexity typically present in robust formulations. Extensive Monte Carlo simulations demonstrate that the approach outperforms conventional estimation methods in terms of bias and mean squared error, as well as in estimating the incubation period. Moreover, applications to real-world surveillance data on COVID-19 illustrate the practical advantages of the proposed method. These findings highlight the method’s robustness and efficiency in scenarios where data contamination from secondary or tertiary infections is common, showing its potential value for early outbreak detection and rapid epidemiological response. I have the following minor revisions for the authors.
- The authors must connect their results to the biological evolution of the COVID-19 pandemic. This area needs to be detailed in the manuscript.
- The code for the analysis cannot be found in the link provided. The authors should provide a direct link to their code for reproducibility.
- In line 38, the authors talked about figure 2, which is later in the article. It is better to make this figure be Figure 1 and bring it to the introduction section for flow.
- Put the figures and tables where they are supposed to be in the text for flow.
Round 2
Reviewer 1 Report
Comments and Suggestions for Authors
Authors properly addressed all concerns. Thanks.